# The Positive Effects of Humic/Fulvic Acid Fertilizers on the Quality of Lemon Fruits

Xiaoying He [1], Hanqi Zhang [1], Jinxue Li [2], Fan Yang [2], Weifeng Dai [1], Cheng Xiang [1] and Mi Zhang [1,*]

1   Faculty of Life Science and Technology, Kunming University of Science and Technology,
    Kunming 650500, China
2   Institute of Tropical and Subtropical Cash Crops, Yunnan Academy of Agricultural Sciences,
    Ruili 678600, China
*   Correspondence: mizhangkmust@126.com; Tel.:+86-871-65920738

**Abstract:** Humic acid (HA) is a kind of organic substance that has shown good effects in regard to promoting crop growth. In the current study, the influences of three kinds of fertilizers mainly containing humic/fulvic acids, including water-soluble fertilizer containing humic acid (WHA), fulvic acid potassium (FAP), and fulvic acid distillate (FAD), on the qualities of 'Eureka' lemon fruits were investigated systematically at their different harvest times. As demonstrated by the results, the indexes used to assess the quality of lemon fruit showed that all lemons treated with those three humic/fulvic acid fertilizers at different harvest times improved in quality. Notably, the single fruit weight, edible rate, and juice yield of lemon fruit in the WHA- and FAP-treated groups, as well as the contents of vitamin C, total acid, total sugar, and total soluble solid, were higher than those in the blank control (CK) group ($p < 0.05$). Furthermore, the contents of total flavonoids and phenols in the peels, pulps, and seeds of lemons in humic/fulvic acid fertilizer–treated groups were all higher than those in the CK group at the three harvest times ($p < 0.05$). This result indicated that humic/fulvic acid fertilizers produced positive effects on the quality of lemon fruits and could be used in lemon planting to improve the quality and added value of lemons.

**Keywords:** humic/fulvic acid fertilizer; lemon fruit; fruit quality; vitamin C; total flavonoids





## 1. Introduction

Humic acid (HA) is a kind of organic substance that widely exists in soil, waters, weathered coal, peat, and lignite [1,2]. On the basis of molecular weight and differences of solubility in different solvents, humic acid could be divided into black humic acid, hymatomelanic acid, and fulvic acid (FA) [3]. So far, humic acid has been used in many fields, especially in agriculture. A series of fertilizers mainly containing humic or fulvic acids have been regarded as green ecological organic fertilizers with the characteristics of fast nutrient supplementation, fast absorption, and a high utilization ratio [4,5]. Additionally, those fertilizers have been found to have effects on increasing crop yield; stimulating plant growth, nutrient uptake, and metabolism [6–11]; regulating soil pH value; increasing soil microbial activity; preventing soil compaction; and reducing heavy-metal pollution [12–15]. However, the influence of humic/fulvic acid fertilizers on the fruit quality, especially internal quality, is still limited. Some previous investigations have shown that spraying with humic acid could increase the average fruit weight, yield, and soluble sugar content of red-flesh and white-flesh pomelo cv. 'Guanximiyou' [16] and grapes [17].

*Citrus limon* (L.) Burm. f. is a small evergreen tree belonging to *Citrus*, family Rutaceae, usually growing in the United States, Italy, Spain, and Southeast Asia. To date, more than 200 cultivars of *C. limon* have been planted in different areas of the world. Many investigations have shown that the lemon fruit is rich in a variety of nutrients and many active ingredients, which can produce antioxidation and antibacterial effects, enhance immunity, and so on [18]. Thus, the lemon has already become one of the representative

functional foods and has obtained many people's favor. In China, the Eureka lemon is one of the most widely grown cultivars in Yunnan, Sichuan, Chongqing, and Hainan, and in those areas, lemon planting has become one of the important local economic industries. However, its development still faces many problems, including the low utilization rate of fertilizers and soil damage caused by the misuse of fertilizers and difference in quality [19]. In view of the characteristics and advantages of humic/fulvic acid fertilizers, we carried out a field experiment to examine the effects of three types of humic/fulvic acid fertilizers, including water-soluble fertilizer containing humic acid (WHA), fulvic acid potassium (FAP), and fulvic acid distillate (FAD), on the qualities of Eureka lemon fruit at different harvest periods in order to explore the application of humic/fulvic acid organic fertilizer in lemon cultivation.

## 2. Materials and Methods

### 2.1. Experimental Materials

The experimental cultivar used was an 8-year-old Eureka lemon tree with a height of 2.4 m and a canopy width of 2.6 m. The field experiment was conducted at the lemon base of the Institute of Tropical and Subtropical Cash Crops, Yunnan Academy of Agricultural Sciences. The soil conditions at that lemon base are yellow or red, with a pH value 5.62. The contents of nitrogen, phosphorus, and potassium were 248.00, 22.18, and 195.18 mg/kg, respectively, and the content of organic matter was 2.09%.

Humic/fulvic acid fertilizers were obtained from Shangcheng Biotechnology Co., Ltd. (Yuxi, China). The labels of those products displayed the following: water-soluble fertilizer containing humic acid (WHA) contains humic acid $\geq$ 30 g/L, $N + P_2O_5 + K_2O \geq$ 200 g/L, and water-insoluble matter $\leq$ 50 g/L; fulvic acid potassium (FAP) contains fulvic acid $\geq$15% and $K_2O \geq$ 15%; and fulvic acid distillate (FAD) contains coal-based fulvic acid $\geq$ 4%.

### 2.2. Experimental Method

The fertilizer-treated method consisted of spraying the lemon tree was with a dilution of humic/fulvic acid fertilizer according to the product manual. Each experiment included three fertilizer-treated groups and one blank control (CK) group, and each group included 4 trees. In the treated groups, WAH was diluted 500 times with a total of 5 L/time/tree, FAP was 2000 times with a total of 5 L/time/tree, and FAD was 500 times with a total of 5 L/time/tree. In the CK group, equivalent distilled water (a total of 5 L/time/tree) was used to spray. The treated time in the experimental period was 29 June 2016, 6 July 2016, and 13 July 2016, respectively. The lemon fruit was picked randomly on 26 December 2016, 26 April 2017, and 26 August 2017, respectively, from the upper, lower, east, south, west, and north directions of each tree. Thirty-six fruits were collected totally from each tree, and each experiment was repeated 3 times.

### 2.3. Testing Index and Method

The single fruit weight was measured by using an electronic scale.

The fruit shape index was calculated according to the following formula: fruit shape index = vertical diameter/horizontal diameter. Moreover, the vertical and horizontal diameter of each fruit was measured by a vernier caliper.

The edible rate was calculated by the following formula: edible rate/% = pulp weight/whole fruit weight $\times$ 100%. The whole-fruit weight was determined by using an electronic scale. The pulp weight was expressed as the weight of the lemon flesh, which was determined by using an electronic scale after the lemon peel was removed.

The juice yield was calculated by the following formula: juice yield/% = juice weight/pulp weight $\times$ 100%. The pulp weight was determined by using an electronic scale after the lemon peel was removed. The juice was obtained by juicing pulp and filtering with 4 layers of gauze and then by the hand-assisted squeezing of pulp again until no juice came out. Finally, it was weighted by using an electronic scale.

The vitamin C content was determined by using 2,6-dichlorophenol indophenol colorimetry [20].

The total acid content was determined by acid–base titration according to the state standard of General Administration of Quality Supervision, Inspection and Quarantine (AQSIQ) of the People's Republic of China (No. GB/T 12456-2008).

The total sugar content was determined by sulfuric acid–anthrone colorimetry [21].

The soluble solid content was determined by Abbe refractometer, according to the ministerial standard of the Ministry of Agriculture of the People's Republic of China (No. NY/T 2637-2014).

The total flavonoids and total phenols were established by NaNO2-Al(NO3)3-NaOH and Folin–Ciocâlteu spectrophotometry, respectively, based on the reported methods [22,23]. Before the determination, the pulp, peel, and seed of each fresh lemon were separated and freeze-dried until the moisture content in these parts was less than 5%. Then each 10 g of pulp, peel, or seed was defatted with 200 mL petroleum ether, using ultrasonic wave for 40 min (repeated for 3 times). After the petroleum ether volatilized, the defatted pulp, peel, or seed was extracted with 200 mL 80% methanol for 40 min (repeated for 2 times), and then the extraction was concentrated to 100 mL and stored at 4 °C. Rutin standard solution (0.2 mg·mL$^{-1}$) and gallic acid standard solution (0.1 mg·mL$^{-1}$) were both prepared with 80% methanol. Rutin standard curve and gallic acid standard curve were established by NaNO$_2$-Al(NO$_3$)$_3$-NaOH and Folin–Ciocâlteu spectrophotometry, respectively. The results showed that a good linear relationship between the absorption value and the rutin concentration was observed when the range of the concentration of rutin was from 10 to 20 mg·L$^{-1}$, in which the linear regression equation was y = 0.0441x − 0.1979, and the correlation coefficient was R$^2$ = 0.9904. In the meantime, the good linear relationship between the gallic acid concentration and its absorbance was observed when it was in the range from 1 to 10 mg·L$^{-1}$, in which the linear regression equation was y = 0.0905x − 0.0032, and the correlation coefficient was R$^2$ = 0.9914.

### 2.4. Methodological Evaluation

A methodological evaluation of the determination of total flavonoids and total phenols was carried out by the experiments of stability, precision, repeatability, and recovery.

In the stability experiment, 1.00 mL of the abovementioned stock solution of lemon peel, pulp, or seed was measured at 0 h, 2 h, 4 h, 8 h, 12 h, 16 h, 20 h, and 24 h, respectively, to obtain the each absorption value based on the corresponding methods of NaNO$_2$-Al(NO$_3$)$_3$-NaOH or Folin–Ciocâlteu spectrophotometry [24,25].

In the precision experiment, 6 parts of 1 mL rutin standard solution and 6 parts of 1 mL gallic acid standard solution were prepared, respectively, whose absorption values were determined according to the standard curve method. The relative standard deviation (RSD) was calculated with the absorbance of total flavonoids or total phenols as the index to investigate the method of precision.

In the repeatability test, we determined the absorbance values of 6 stock solutions of lemon peel, pulp, or seed by using the standard curve drawing method. The RSD was calculated by the absorbance of total flavonoids and total phenols as the index, which was used to investigate the repeatability of this method.

In the recovery test, 9 samples of lemon peel, pulp, or seed with known content were divided into 3 groups. Then 0.8, 1, and 1.2 times of rutin or gallic acid reference solution were added into them, respectively. The recovery rate was evaluated by using the RSD values calculated by the content of total flavonoids and total phenols that was obtained by the standard curve drawing method and the average absorbance value.

### 2.5. Data Analysis

Experimental data were processed by EXCEL 2003 software. A statistical analysis was performed by SPSS 20.0 and R software 4.2.0.

## 3. Results

### 3.1. Effects of Humic/Fulvic Acid Fertilizer on the Single Fruit Weight of Lemon

Compared with the CK group, the weight of a single lemon fruit in each treated group was significantly increased ($p < 0.05$) (Table 1). Moreover, the weights of a single lemon fruit in the WHA-treated groups on 26 April 2017 and 26 August 2017 were higher than they were for the other two groups; meanwhile, in the FAP-treated group, the weights were higher on 26 December 2016. However, the weight increase of a single lemon fruit in the FAD-treated group at the three harvest times was the lowest among the three treated groups.

### 3.2. Effects of Humic/Fulvic Acid Fertilizer on the Shape Index of Lemon Fruit

At the tested harvest time, the shape index of the lemon fruits in each group was within the range from 1.34 to 1.39, including the treated group and CK group (Table 1). The statistical analysis showed that there was no significant difference in the shape index of lemon fruits among the three treated groups and the CK group at each harvest time.

### 3.3. Effects of Humic/Fulvic Acid Fertilizer on the Edible Rate of Lemon Fruit

At the three harvest times, the edible rate of lemon fruits in the WHA- and FAP-treated groups showed a significant increase ($p < 0.05$) compared with their corresponding CK groups ($p > 0.05$); meanwhile, in the FAD-treated groups, only on two harvest times of 26 April 2017 and 26 August 2017 was the edible rate obviously increased ($p < 0.05$) compared with the corresponding CK groups ($p < 0.05$) (Table 1). Overall, the edible rate of lemon fruits in those three humic/fulvic acid–treated groups was higher at the harvest time of 26 August 2017 than it was at the other two harvest times.

### 3.4. Effects of Humic/Fulvic Acid Fertilizer on Juice Yield of Lemon Fruit

A comparison of juice yields in three humic/fulvic acid–treated groups at each harvest time with their corresponding CK groups indicated that the humic/fulvic acid fertilizer could increase the yield of lemon fruit juice significantly ($p < 0.05$) (Table 1). Notably, the juice yield of lemon fruits in the WHA- and FAD-treated groups was higher at the harvest time of 26 August 2017 than it was for the other two harvest times; in the FAP-treated groups, it was higher at the harvest time of 26 December 2016 than it was at the other two harvest times.

### 3.5. Effects of Humic/Fulvic Acid Fertilizer on Vitamin C Content of Lemon Fruit

It was observed that the vitamin C content of the lemon fruit in the three humic/fulvic acid–treated groups at each harvest time increased obviously compared with their corresponding CK groups ($p < 0.05$). Additionally, the vitamin C content of the lemon fruit in each treated group was higher at the harvest time of 26 August 2017 than it was for the other two harvest times (Table 1).

### 3.6. Effects of Humic/Fulvic Acid Fertilizer on Total Acid Content of Lemon Fruit

Table 1 shows the total acid content of lemon fruits in each treated group at each harvest time increased obviously ($p < 0.05$) when compared with each corresponding CK group. As with the vitamin C content, the total acid content of the lemon fruit in each treated group was higher at the harvest time of 26 August 2017 than it was for other two harvest times.

### 3.7. Effects of Humic/Fulvic Acid Fertilizer on Total Sugar Content of Lemon Fruit

Compared with the CK group, the total sugar content of single lemon fruit in each treated group was significantly increased ($p < 0.05$). At the same harvest times, including 26 December 2016 and 26 August 2017, the total sugar content in FAP-treated group was higher than that in the other two treated groups. However, at the time of harvest, on 26 April 2017, the content of total sugar in the WHA-treated group was higher than that in the other treated groups (Table 1).

**Table 1.** Effects of humic/fulvic acid fertilizers on quality of lemon fruit.

| Quality Index | 26 December 2016 | | | | 26 April 2017 | | | | 26 August 2017 | | | |
|---|---|---|---|---|---|---|---|---|---|---|---|---|
| | WHA | FAP | FAD | CK | WHA | FAP | FAD | CK | WHA | FAP | FAD | CK |
| Single fruit weight (g) | 111.64 [a] ± 0.32 | 112.29 [a] ± 0.21 | 109.45 [b] ± 0.24 | 106.32 [c] ± 0.26 | 106.37 [a] ± 0.30 | 103.49 [b] ± 0.36 | 102.98 [b] ± 0.37 | 100.40 [c] ± 0.44 | 117.85 [a] ± 0.42 | 115.50 [b] ± 0.50 | 112.94 [c] ± 0.31 | 111.90 [c] ± 0.30 |
| Fruit shape index | 1.36 [a] ± 0.05 | 1.37 [a] ± 0.07 | 1.39 [a] ± 0.04 | 1.36 [a] ± 0.02 | 1.36 [a] ± 0.03 | 1.37 [a] ± 0.06 | 1.38 [a] ± 0.09 | 1.36 [a] ± 0.07 | 1.37 [a] ± 0.04 | 1.35 [a] ± 0.03 | 1.35 [a] ± 0.08 | 1.34 [a] ± 0.06 |
| Edible rate (%) | 72.65 [a] ± 0.60 | 72.70 [a] ± 0.52 | 71.13 [b] ± 0.49 | 69.94 [b] ± 0.61 | 70.01 [a] ± 0.75 | 68.54 [b] ± 0.40 | 68.51 [b] ± 0.51 | 66.08 [c] ± 0.47 | 76.62 [a] ± 0.69 | 74.58 [b] ± 0.75 | 73.99 [b] ± 0.50 | 71.70 [c] ± 0.46 |
| Juice yield (%) | 55.89 [b] ± 0.81 | 57.21 [a] ± 0.60 | 54.90 [b] ± 0.71 | 51.32 [c] ± 0.79 | 53.77 [a] ± 0.59 | 51.03 [b] ± 0.72 | 50.91 [b] ± 0.68 | 48.95 [c] ± 0.60 | 59.27 [a] ± 0.54 | 56.20 [b] ± 0.89 | 55.97 [b] ± 0.61 | 52.46 [c] ± 0.70 |
| Vitamin C (mg/100 g) | 67.10 [b] ± 0.49 | 68.61 [a] ± 0.61 | 66.95 [b] ± 0.78 | 61.85 [c] ± 0.76 | 66.99 [a] ± 0.89 | 65.39 [b] ± 0.74 | 64.06 [c] ± 0.55 | 59.02 [d] ± 0.49 | 72.01 [a] ± 0.60 | 69.17 [b] ± 0.35 | 67.05 [c] ± 0.62 | 63.30 [d] ± 0.70 |
| Total acid (g/kg) | 47.51 [a] ± 0.44 | 47.69 [a] ± 0.58 | 45.05 [b] ± 0.65 | 43.88 [c] ± 0.70 | 45.97 [a] ± 0.51 | 44.10 [b] ± 0.49 | 43.96 [b] ± 0.78 | 42.04 [c] ± 0.56 | 49.56 [a] ± 0.48 | 49.08 [a] ± 0.39 | 48.79 [a] ± 0.60 | 45.82 [b] ± 0.54 |
| Total sugar (g/100g) | 1.92 [b] ± 0.10 | 2.00 [a] ± 0.08 | 1.90 [b] ± 0.12 | 1.70 [c] ± 0.06 | 2.09 [a] ± 0.13 | 2.05 [a] ± 0.09 | 1.94 [b] ± 0.08 | 1.81 [c] ± 0.14 | 1.81 [b] ± 0.11 | 1.97 [a] ± 0.10 | 1.82 [b] ± 0.06 | 1.65 [c] ± 0.05 |
| Total soluble solid (%) | 8.24 [a] ± 0.21 | 8.15 [a] ± 0.16 | 7.94 [b] ± 0.27 | 7.69 [b] ± 0.14 | 8.31 [a] ± 0.29 | 8.30 [a] ± 0.12 | 7.99 [b] ± 0.31 | 7.72 [b] ± 0.20 | 8.20 [a] ± 0.27 | 7.71 [b] ± 0.22 | 7.60 [b] ± 0.20 | 7.58 [b] ± 0.34 |

Dates are expressed as the mean value ± SD (n = 36). [a, b, c] Different small letters in the same row indicate the significant differences among lemon fruits treated with different fertilizers at the same harvest time ($p < 0.05$).

### 3.8. Effects of Humic/Fulvic Acid Fertilizer on Total Soluble Solid Content of Lemon Fruit

For the total soluble solid content of the lemon fruit, a significant increase was only present in the WHA-treated group at each harvest time ($p < 0.05$), compared with the CK group (Table 1). In the FAP-treated group, the total soluble solid content of the lemon fruit significantly increased on 26 December 2016 and 26 April 2017. However, there was no statistical difference in regard to the total soluble solid content between the FAD-treated group and the CK group at each harvest time ($p > 0.05$). Furthermore, the total soluble solid content of the lemon fruit in the WHA-treated group was always higher than that of the other two treated groups at each harvest time.

### 3.9. Effects of Humic/Fulvic Acid Fertilizer on the Contents of Total Flavonoids and Total Phenols of Lemon Fruit

The RSD values determined in the tests for methodological evaluation, including stability, precision, repeatability, and recovery, were all less than 2% (Table 2), thus suggesting that the present method used in the determination of the contents of total flavonoids and total phenols has good stability, precision, repeatability, and recovery.

**Table 2.** The RSD values (%) of methodological evaluation.

| Analyte | Part | Stability | Repeatability | Recovery | Standard | Precision (n = 6) |
|---|---|---|---|---|---|---|
| Total flavonoids | Peel | 0.57 | 0.72 | 0.39 | Rutin | 0.70 |
| | Pulp | 0.64 | 1.03 | 0.80 | | |
| | Seed | 0.63 | 0.65 | 0.41 | | |
| Total phenols | Peel | 0.92 | 0.52 | 0.76 | Gallic acid | 0.32 |
| | Pulp | 0.79 | 0.42 | 0.54 | | |
| | Seed | 0.85 | 0.46 | 0.40 | | |

As shown in Figure 1, the contents of total flavonoids or phenols from the peel, pulp, and seed in all treated groups showed an obvious increase at three collection points, as compared with the CK group ($p < 0.05$). A comparison of the contents among these three treated groups indicated that the total flavonoid or phenol content was higher in the WHA-treated group, followed by the FAP- and FAD-treated groups. Furthermore, the content of total flavonoids or phenols in the peels at different harvests in each group was higher than that in the other two parts of the lemon fruit.

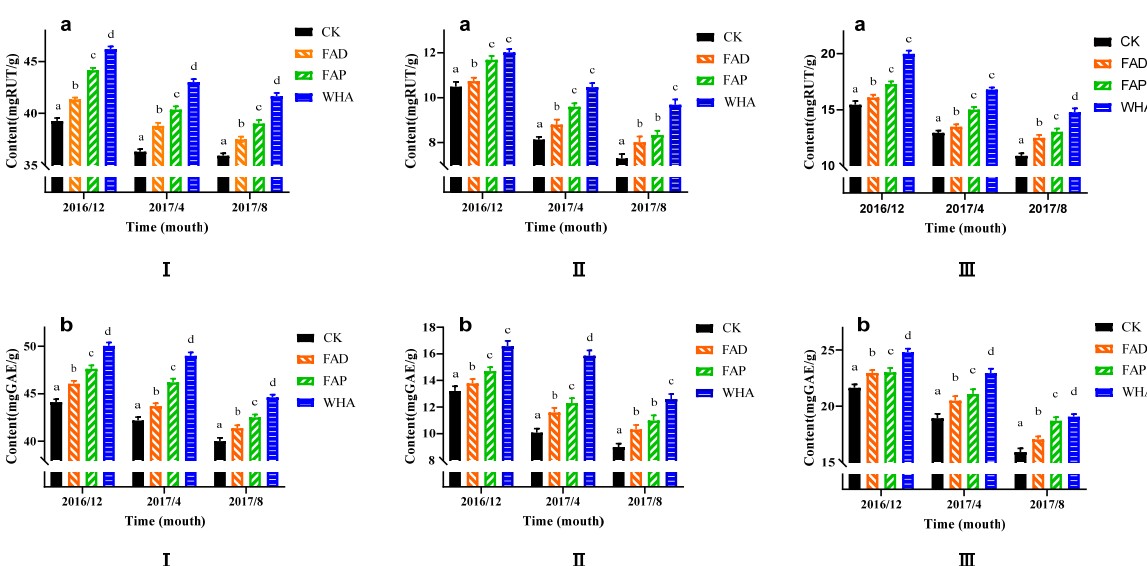

**Figure 1.** The effects of humic/fulvic acid fertilizer on the contents of total flavonoids (**a**) and total phenols (**b**) of lemon fruits. I, peel; II, pulp; III, seed. Different small letters in this figure indicate the significant differences among different treated groups ($p < 0.05$).

## 4. Discussion

The lemon is a kind of subtropical fruit, and a lemon tree can bloom and bear fruit many times throughout the year. It grows very fast and needs a large amount of fertilizer. It is well-known that the quality and application of fertilizer directly impact on crop quality, as well as the soil structure and ecological environment [26,27]. So far, humic/fulvic acid fertilizer has been widely applied to many planted crops and confirmed to produce many good effects on crop quality, as well as on soil structure and ecological environment. However, the research on the systematic evaluation of its specific impact on fruits is very limited.

In our study, four indexes were used to express the external quality of lemon fruit influenced by humic/fulvic acid fertilizer, including single fruit weight, fruit shape index, edible rate, and juice yield. Those indexes are also the most common and direct parameters used for evaluating the fruit quality. The results showed that all of the indexes in the different treated groups at three harvest times increased compared with their corresponding CK groups, except for fruit shape indexes between the treated groups and CK groups, as they had no statistical difference (Figure 2). These results indicate that humic/fulvic acid fertilizer could produce positive effects on the external quality of lemon fruit.

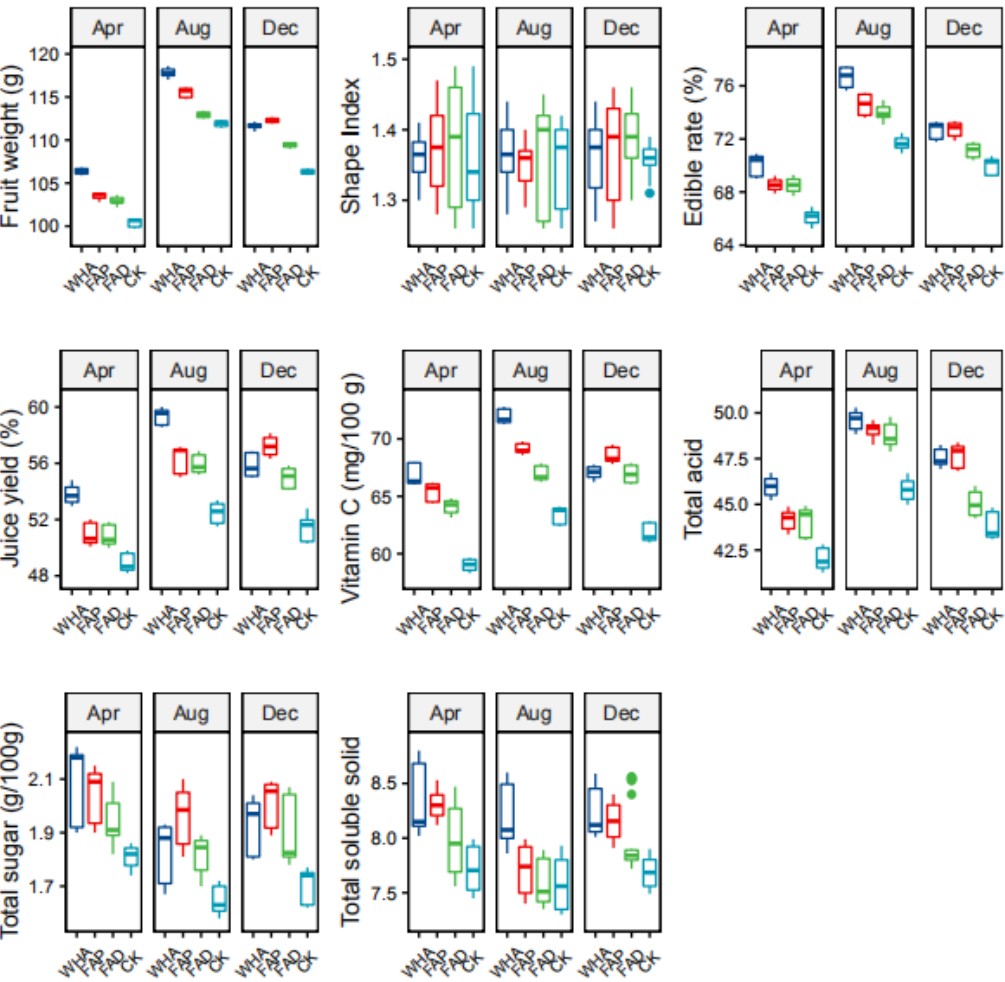

**Figure 2.** Analysis of effects of humic/fulvic acid fertilizer on quality of lemon fruit presented by boxplot.

It is well-known that lemon fruits are not only for eating, but also can be made into some industrial materials and medicines. The peel of a lemon is rich in essential oil, flavonoids, and phenols, which can be used as additives in some cosmetics, food, and drugs. The pulp of the lemon can be used to produce pectin, which is an important material

for the production of candy, candied fruit, jam, and health products. Those essential oil, flavonoids, phenols, and pectin are the chemical constituents in lemon that cannot be directly evaluated by intuitive knowledge or visual feeling but can bring more added value to the lemon fruit commercially. Thus, the content of chemical constituents in the lemon can represent the internal quality of lemon fruit.

In our present research, the contents of vitamin C, total acid, total sugar, total soluble solid, total flavonoids, and total phenols were detected to evaluate the effects of humic/fulvic fertilizer on the internal quality of lemon fruit in three harvest times. As a result, those six indexes in the treated groups all showed an uptrend compared with the corresponding CK groups, with the exception that the content of total soluble solids in the FDA-treated group has no statistical difference from that in the CK groups (Figures 1 and 2). However, it still revealed that the effects of humic/fulvic acid fertilizer on the internal quality of lemon fruit is positive, and this can help improve lemons' added value.

Vitamin C, organic acid, sugar, flavonoids, and phenols are the plant metabolites, whose contents are regulated by the plant genetic and environmental factors [28–30]. Some research has shown that humic acid substances, as an environmental factor, can affect plant metabolism by regulating the plant protease activity or inducing protease synthesis [31–33]. Therefore, it is speculated that humic/fulvic acid fertilizers in our study promoting these metabolite contents in lemons were related to their effects on some proteases.

## 5. Conclusions

Overall, humic/fulvic acid fertilizer, including water-soluble fertilizer containing humic acid (WHA), fulvic acid potassium (FAP), and fulvic acid distillate (FAD), can improve the quality of lemon fruits to varying degrees. WHA and FAP can significantly increase the single fruit quality, edible rate, juice yield, contents of vitamin C, total acid, total sugar, and soluble solid of lemon fruit, but they have little influence on the fruit shape index of lemon.

**Author Contributions:** Conceptualization, M.Z.; methodology, X.H., H.Z., J.L. and F.Y.; validation, W.D. and C.X.; formal analysis, H.Z.; investigation, X.H.; resources, J.L. and F.Y.; data curation, X.H. and H.Z.; writing—original draft preparation, X.H. and H.Z.; writing—review and editing, M.Z.; visualization, X.H. and H.Z.; supervision, W.D. and C.X.; project administration, M.Z.; funding acquisition, M.Z. All authors have read and agreed to the published version of the manuscript.

**Funding:** This research was funded by the Applied Basic Research Foundation of Yunnan Province (202101AT070133) and the National Natural Science Foundation of China (No. 21466018).

**Institutional Review Board Statement:** Not applicable.

**Informed Consent Statement:** Not applicable.

**Data Availability Statement:** Not applicable.

**Conflicts of Interest:** The authors declare no conflict of interest.

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
