# Peer review of "The Positive Effects of Humic/Fulvic Acid Fertilizers on the Quality of Lemon Fruits"

_agronomy, doi:10.3390/agronomy12081919_

Round 1

Reviewer 1 Report

Congrats for your scientific paper! It is very interesting and maybe you will can extend your studies to orange and grapefruit too.

Author Response

Thanks very much for your valuable comment. More research about other fruits, such as orange, grape, as your suggestion, will be carried out by our group in the future.

Reviewer 2 Report

1.    In this study, the physico-chemical properties of the soil used for growing lemons were not explained, the information available was too little, only pH and NPK levels, so that other growth inhibiting factors were unknown.

2.    In your research methodology you do not use basic fertilizers but only rely on humic/fulvic fertilizers, even though the original NPK content of the soil is relatively low. There is no information on the levels of other nutrients such as Ca, Mg and micronutrients in the soil. Are you sure that citrus plants are not deficient in these nutrients?

3.    In my opinion, the data presented in tabular form is the same as that presented in graph form, so it is redundant. This should be avoided as it is just repetition of data.

4. The discussion is too simple, there is no explanation of how the mechanism of the role of humic/fulvic acid in improving the quality of lemons.

Author Response

We deeply appreciate your comments (agronomy-1847634). The following is our response.

1.In this study, the physico-chemical properties of the soil used for growing lemons were not explained, the information available was too little, only pH and NPK levels, so that other growth inhibiting factors were unknown.

Response: Thanks for your comment. We totally agreed with you that the physico-chemical properties of the soil have effects on the quality of fruits.  In order to avoid the influence of factors from soil, we selected the spraying method to treat the lemon tree and same condition of soil to culture the tree. Further, the difference between CK group and humic/fulvic fertilizer- treated group was only water treatment or humic/fulvic fertilizer treatment in each experiment, which could help us to clarify the effects of humic/fulvic fertilizers on the quality of lemons.

2.In your research methodology you do not use basic fertilizers but only rely on humic/fulvic fertilizers, even though the original NPK content of the soil is relatively low. There is no information on the levels of other nutrients such as Ca, Mg and micronutrients in the soil. Are you sure that citrus plants are not deficient in these  nutrients?

Response: Thanks for your comment. As far as we know, Citrus plants require many elements for growth, development and fruit formation, including N, P, K, Ca, Mg, S, B, Zn, Fe, Cu, Mn, Mo, and so on. Humic/fulvic fertilizers used in our study are the commercial products, which contain humic/fulvic acids and other nutrients. But the advantage of humic/fulvic fertilizers is high contents of humic/fulvic acids in them. And the aim of our study is to evaluate the effects of humic/fulvic fertilizers on the quality of lemons.

3. In my opinion, the data presented in tabular form is the same as that presented in graph form, so it is redundant. This should be avoided as it is just repetition of data.

Response: Thanks for your comment. The data in boxplot (Figure 2) were the median, which could reflect the distribution probability of the samples. And the data in table 1 were the means ± SD, which reflect the central tendency of the data. So please consider keeping them.  

 4.The discussion is too simple, there is no explanation of how the mechanism of the role of humic/fulvic acid in improving the quality of lemons.

Response: Thanks for your comments. Based on our current data and reference, we have added more discussion about the mechanism of the role of humic/fulvic acid in improving the quality of lemons in our discussion part.

Reviewer 3 Report

No discussion of results with the literature. An introduction to the effects of humic and fulvic acids on fruit yield and quality should also be developed. Cite more positions of  literature. The stimulating effect of humic substances on plants should be mentioned.

Author Response

Thanks for your comments, based on which, we have revised our introduction and discussion part in our revised MS.